# Carbapenemase-Producing *Klebsiella pneumoniae* in COVID-19 Intensive Care Patients: Identification of IncL-VIM-1 Plasmid in Previously Non-Predominant Sequence Types

**DOI:** 10.3390/antibiotics12010107

**Published:** 2023-01-06

**Authors:** Javier E. Cañada-García, Eva Ramírez de Arellano, Miguel Jiménez-Orellana, Esther Viedma, Aida Sánchez, Almudena Alhambra, Jennifer Villa, Alberto Delgado-Iribarren, Verónica Bautista, Noelia Lara, Silvia García-Cobos, Belén Aracil, Emilia Cercenado, María Pérez-Vázquez, Jesús Oteo-Iglesias

**Affiliations:** 1Laboratorio de Referencia e Investigación en Resistencia a Antibióticos e Infecciones Relacionadas con la Asistencia Sanitaria, Centro Nacional de Microbiología, Instituto de Salud Carlos III, Majadahonda, 28222 Madrid, Spain; 2CIBER de Enfermedades Infecciosas (CIBERINFEC), Red Española de Investigación en Patología Infecciosa (REIPI), Instituto de Salud Carlos III, 28029 Madrid, Spain; 3Servicio de Microbiología, Hospital Universitario 12 de Octubre, Instituto de Investigación Hospital 12 de Octubre (imas12), 28041 Madrid, Spain; 4Laboratorio de Microbiología, URSalud UTE, Hospital Infanta Sofía, San Sebastián de los Reyes, 28702 Madrid, Spain; 5Servicio de Microbiología, Laboratorios ABACID, HM Hospitales, 28050 Madrid, Spain; 6Servicio de Microbiología Clínica, Hospital Clínico San Carlos, 28040 Madrid, Spain; 7Servicio de Microbiología, Hospital General Universitario Gregorio Marañón, 28007 Madrid, Spain; 8CIBER de Enfermedades Respiratorias (CIBERES), Instituto de Salud Carlos III, 28029 Madrid, Spain

**Keywords:** *Klebsiella pneumoniae*, carbapenemases, COVID-19, antibiotic resistance, intensive care units (ICUs), outbreaks, cgMLST, WGS

## Abstract

During the COVID-19 pandemic, intensive care units (ICUs) operated at or above capacity, and the number of ICU patients coinfected by nosocomial microorganisms increased. Here, we characterize the population structure and resistance mechanisms of carbapenemase-producing *Klebsiella pneumoniae* (CP-Kpn) from COVID-19 ICU patients and compare them to pre-pandemic populations of CP-Kpn. We analyzed 84 CP-Kpn isolates obtained during the pandemic and 74 CP-Kpn isolates obtained during the pre-pandemic period (2019) by whole genome sequencing, core genome multilocus sequence typing, plasmid reconstruction, and antibiotic susceptibility tests. More CP-Kpn COVID-19 isolates produced OXA-48 (60/84, 71.4%) and VIM-1 (18/84, 21.4%) than KPC (8/84, 9.5%). Fewer pre-pandemic CP-Kpn isolates produced VIM-1 (7/74, 9.5%). Cefiderocol (97.3–100%) and plazomicin (97.5–100%) had the highest antibiotic activity against pandemic and pre-pandemic isolates. Sequence type 307 (ST307) was the most widely distributed ST in both groups. VIM-1-producing isolates belonging to ST307, ST17, ST321 and ST485, (STs infrequently associated to VIM-1) were detected during the COVID-19 period. Class 1 integron Int1-*bla*_VIM-1_-*aac*(6*′*)-1*b*-*dfrB*1-*aadA*I-*catB*2-*qacE*Δ1/*sul*1, found on an IncL plasmid of approximately 70,000 bp, carried *bla*_VIM-1_ in ST307, ST17, ST485, and ST321 isolates. Thus, CP-Kpn populations from pandemic and pre-pandemic periods have similarities. However, VIM-1 isolates associated with atypical STs increased during the pandemic, which warrants additional monitoring and surveillance.

## 1. Introduction

Coronavirus disease of 2019 (COVID-19), the disease caused by infection with the SARS-CoV-2 virus, has caused the worst pandemic since the 1918 influenza. Through the end of October 2022, 627 million confirmed cases and 6.5 million deaths from COVID-19 have been reported globally [1]. Patients infected by SARS-CoV-2 can be coinfected with *Klebsiella pneumoniae* while in the intensive care unit (ICU). During the pandemic, the capacity of many ICUs was exceeded due to the large increase in hospital admissions, which could have increased the frequency of co-infection with nosocomial microorganisms such as carbapenemase-producing Enterobacterales (CPE). A recent study in China showed that 94.2% of COVID-19 patients were coinfected by one or more pathogens and that *K. pneumoniae* was the second most frequently detected co-infecting microorganism [2].

Currently, healthcare-related infections are a major public health problem worldwide and the frequency of infection by multidrug-resistant (MDR) bacteria is increasing. This threat is especially relevant in ICUs, where patients are at higher risk for nosocomial infections mainly due to the presence of MDR microorganisms in the hospital environment [3]. 

CPE infections, especially carbapenemase-producing *K. pneumoniae* (CP-Kpn), are one of the most significant threats to public health [4] because the carbapenems are among the last available therapeutic drugs for eradicating these infections. In addition, CP-Kpn isolates frequently demonstrate increased capacity for persistence, rapid dispersion, and emergence, which has implications for colonization, spread of high-risk MDR clones, and co-resistance to non-carbapenem antibiotics [5]. Thus, CP-Kpn bacterial strains (mainly producing OXA-48, KPC, NDM, and VIM) are a significant public health problem in Spain and other European countries [6].

The emergence of COVID-19 and its spread as a global pandemic has modified some of the factors that favor spread of MDR pathogens [7]. Some of these factors favor dissemination of MDR microorganisms (i.e., increased antibiotic consumption, higher occupancy of ICUs, longer hospital stays, and more frequent endotracheal intubation of patients), but other factors could prevent such dissemination (i.e., improved personal hygiene and adoption of personal protective equipment). The impact of these factors on the frequency of infection with antibiotic-resistant bacteria has been evaluated in other countries [8,9].

The public health significance of CPE in Spain has been studied [5]. However, the degree to which the COVID-19 pandemic has changed CPE status remains to be determined. This could be important because the COVID-19 pandemic could have qualitatively and quantitatively increased the threat of CPE and reduced the overall efficacy of antibiotic therapy. The aim of this research project was (i) to characterize CP-Kpn isolates and sequence types (STs) in COVID-19 patients in the ICU and (ii) to compare the epidemiological and microbiological characteristics of CP-Kpn isolates obtained during the COVID-19 pandemic with those obtained before the onset of the pandemic.

## 2. Results

### 2.1. Bacterial Isolates, Patients, and Carbapenemase Types

This study compares CP-Kpn isolates obtained during the COVID-19 pandemic with isolates obtained before the pandemic began in 2019. The analysis included 85 non-duplicate CP-Kpn isolates from COVID-19 patients admitted to ICUs in five hospitals in Madrid. Most of these pandemic-period isolates were from men (71.8%) and people under 65 years old (56.5%). Of the 85 isolates, 49 (57.7%) caused clinical infections, including 25 (29.4%) urinary tract infections, six (7.1%) respiratory tract infections, eight (9.4%) bacteremia cases, and 10 (11.8%) other infections; the remaining 36 (42.3%) were isolated from rectal samples.

Whole genome sequencing (WGS) identified carbapenemase genes in the DNA of pandemic-period isolates of CP-Kpn, including 60 isolates (71.4%) carrying *bla*_OXA-48_, 18 (21.4%) carrying *bla*_VIM-1_, four (4.8%) carrying *bla*_KPC-2_, and four (4.8%) carrying *bla*_KPC-3_. Two isolates had both *bla*_OXA-48_ and *bla*_KPC-2_ genes. One isolate was excluded from subsequent molecular analyses because WGS data identified it as a *Klebsiella variicola* variant.

In addition, 34 pre-pandemic CP-Kpn isolates collected in Madrid hospitals as part of the CARB-ES-19 project [5] were included in this study (Madrid-CARB-ES-19). Most of the pre-pandemic isolates were from women (55.9%) and half were from individuals more than 65 years old. Of the 34 isolates, all produced clinical infections, including 16 (47.1%) urinary tract infections, four (11.8%) respiratory tract infections, seven (20.6%) bacteremia cases, and seven (20.6%) other infections. Twenty-five (73.5%) were OXA-48-producers, and 10 (29.4%) were KPC-producers (nine KPC-3-producers and one KPC-2-producer). One isolate co-produced OXA-48 and KPC-2.

A second group of pre-pandemic samples collected by the CARB-ES-19 project included 40 isolates of CP-Kpn from patients in the ICU (ICU-CARB-ES-19) [5]. Most of these isolates were from men (55%), and 30% were from individuals more than 65 years old. All isolates produced clinical infections, including seven (17.5%) urinary tract infections, 14 (35%) respiratory tract infections, 10 (25%) bacteremia cases, and nine (22.5%) other infections. Twenty-seven (67.5%) carried *bla*_OXA-48_, 7 (17.5%) carried *bla*_VIM-1_, five (12.5%) carried *bla*_KPC_ (four *bla*_KPC-3_ and one *bla*_KPC-2_), and one (2.5%) carried *bla*_IMP-8_.

### 2.2. Antibiotic Susceptibility Testing

The in vitro antibiotic susceptibilities of the pandemic-period isolates of CP-Kpn (COVID, *n* = 85) and two groups of pre-pandemic isolates of CP-Kpn (Madrid-CARB-ES-19, *n* = 34 and ICU-CARB-ES-19, *n* = 40 isolates, respectively) were analyzed and the results are listed in Table 1. The percent of isolates susceptible to the following carbapenems were 41.2%, 52.9%, and 57.5% susceptible to imipenem; 37.6%, 52.9%, and 50% susceptible to meropenem; and 2.4%, 0%, and 2.5% susceptible to ertapenem, respectively, with 98.7% of all isolates not susceptible to at least one carbapenem (Table 1).

The antibiotics with the highest activity against CP-Kpn isolates in COVID-19, Madrid-CARB-ES-19 and ICUs-CARB-ES-19 groups were, respectively, cefiderocol (98.8%, 97.3%, and 100%), plazomicin (97.6%, 100%, and 97.5%), meropenem/vaborbactam (87.1%, 82.4%, and 92.5%), colistin (85.9%, 91.2%, and 85%), imipenem/relebactam (82.4%, 73.5%, and 77.5%), and ceftazidime/avibactam (75.3%, 100%, and 72.5%; Table 1). However, these percentages varied depending on the carbapenemase type in CP-Kpn COVID-19 (Table 2) isolates as previously described in the CARB-ES-19 study [5].

### 2.3. Phylogenetic Analysis of CP-Kpn

The STs of the 84 pandemic-period CP-Kpn isolates were determined by WGS. The results identified 20 STs (Table 3), with the most prevalent STs being represented by ≥4 isolates including ST307 (20; 23.8%), ST15 (15; 17.8%), ST11 (14; 16.7%), ST17 (8; 9.5%), and ST485 (4; 4.8%), comprising 72.6% of all isolates. ST307 was represented in isolates from all five hospitals participating in this study. A total of 23 (27.4%) CP-Kpn isolates belonged to minority STs (STs with less than four isolates), but these STs corresponded to 75% of the total STs detected in the study (15/20). The SDI was 23.8. ST307 expressed four distinct carbapenemases (KPC-2, KPC-3, OXA-48, and VIM-1), whereas ST11 and ST15 only expressed OXA-48 and ST485 only expressed VIM-1. All but two of the ST17 isolates produced VIM-1 (Table 3).

The 34 pre-pandemic Madrid-CARB-ES-19 CP-Kpn isolates belonged to nine STs with an SDI of 26.5. The most prevalent STs were ST307 (13; 38.2%), ST11 (6; 17.6%), and ST15 (5; 14.7%), corresponding to 70.6% of these isolates (Table 3). ST307 expressed carbapenemases KPC-3 and OXA-48, none of the isolates produced VIM-1, and all ST11 and ST15 isolates produced OXA-48 (Table 3).

The 40 pre-pandemic ICU-CARB-ES-19 CP-Kpn isolates belonged to 18 STs (Table 2) with an SDI of 45. The most prevalent STs were ST307 (9; 22.5%), ST11 (7; 17.5%), and ST147 (4; 10%), corresponding to 50% of these isolates (Table 3). The majority of the ST307 and ST11 isolates produced OXA-48, and all ST147 isolates produced VIM-1 (Table 3).

A minimum-spanning tree was constructed for all 84 pandemic-period CP-Kpn isolates using the gene-by-gene approach, with allelic distance calculated using cgMLST (Figure 1), excluding clusters with three or fewer isolates. The average allelic distances between pairs of isolates by cluster were 39 alleles in Cluster 1 (ST307) (range: 0–86), 158 alleles in Cluster 2 (CC11) (range: 0–572), 42 alleles in Cluster 3 (ST15) (range: 0–577), 281 alleles in Cluster 4 (ST17) (range: 0–700), and 2 alleles in Cluster 5 (ST485) (range: 0-3). Applying a relatedness threshold of 10 alleles, five groups with more than three related isolates were detected. Two of these groups included ST307 isolates, the first with three isolates producing VIM-1 and five isolates producing OXA-48 and the second group including four isolates producing KPC-3. One group included six VIM-1-producing ST17 isolates, another included five OXA-48-producing ST11 isolates, and the fifth group included four VIM-1-producing ST485 isolates (Figure 1).

We then compared pandemic-period COVID CP-Kpn isolates with Madrid-CARB-ES-19 and ICU-CARB-ES-19 isolates, as described above. Five clusters included isolates from all three study groups; these clusters included ST307, CC11, ST15, ST147, and ST392 isolates. One additional cluster included eight ST17 isolates from the pandemic period and one isolate from the ICU-CARB-ES-19 group, and other cluster included four ST485 isolates from pandemic-period COVID-19 patients (Figure 2).

Applying a threshold of 10 alleles [10], we identified four groups (*n* > 3 isolates per group) of related isolates from pandemic-period COVID-19 and pre-pandemic CARB-ES-19 groups: two groups in the CC11 cluster with OXA-48-producers, one group in the ST307 cluster included KPC-3-producers, and one group in the ST15 cluster included OXA-48-producers (Figure 2). Three groups were identified that included only pandemic-period isolates from COVID-19 patients; one of these groups included eight ST307 isolates that produced OXA-48 and VIM-1, whereas the other two groups included four ST485 and six ST17 VIM-1-producers, respectively (Figure 2).

Comparing all VIM-1-producing isolates, 18/84 pandemic-period isolates from five STs expressed VIM-1, whereas 7/74 pre-pandemic isolates from four STs expressed VIM-1. The predominant STs among the VIM-1-producing pandemic period isolates were ST307, ST17, ST485, and ST321 (Figure 3).

### 2.4. Identification and Distribution of Resistance and Virulence Genes

An average of 11 acquired resistance genes (ARGs) were detected in COVID-19 isolates (range: 2–18 ARGs), 11.6 in Madrid-CARB-ES-19 isolates (range: 2–16 ARGs), and 9.9 in ICU-CARB-ES-19 isolates (range: 2–18 ARGs) (Appendix A). 

The most frequently identified extended-spectrum β-lactamase (ESBL) gene was *bla*_CTX-M-15_. This gene was detected in 75% of COVID-19 CP-Kpn isolates, 76.5% of Madrid-CARB-ES-19 isolates, and 50% of CP-Kpn ICU-CARB-ES-19 isolates. Other ESBL genes detected were *bla*_CTX-M-9_ (2.4% in COVID-19 group, 5% in ICU-CARB-ES-19 group, and not detected in Madrid-CARB-ES-19 group), *bla*_SHV-12_ (2.4% in COVID-19 group, 5% in ICU-CARB-ES-19 group, and not detected in Madrid-CARB-ES-19 group), *bla*_SHV-2_ (only detected in one isolate of COVID-19 group), and *bla*_SHV-65_ (only detected in one isolate of ICU-CARB-ES-19 group) (Table 4).

The predominant aminoglycoside resistance genes encoded N-acetyltransferases, including *aac*(6*′*)-*Ib*-*cr* (50%, 64.7%, and 45% in COVID-19, Madrid-CARB-ES-19, and ICU-CARB-ES-19 groups, respectively), *aac*(3)-*IIa* (33.3%, 58.8%, and 42.5%, respectively), *aac*(6*′*)-*Ib* (19%, 5.9%, and 20%, respectively), and *aph*(3``)-*Ib/aph* (6)-*Id* (53.6%, 67.6%, and 40%, respectively) (Table 4).

ARGs encoding resistance to chloramphenicol, sulfonamides, trimethoprim, and tetracyclines were analyzed in the three groups of patients (Table 4 and Appendix A). Chloramphenicol resistance genes were less frequent in pandemic isolates [40.5%, mainly *catA*1 (22.6%) and *catB*2 (20.2%)] than in pre-pandemic isolates [70.5% and 70%, mainly *catB*3 (64.71% and 42.5%) in Madrid-CARB-ES-19 and ICU-CARB-ES-19 groups, respectively].

Four isolates contained *mcr-9*, two in COVID-19 and ICU-CARB-ES-19 groups although only one was resistant to colistin (Table 4 and Appendix A).

Plasmid-mediated quinolone resistance *qnr*-like determinants were detected in 38 (45.2%), 21 (61.8%), and 20 (50%) isolates of COVID-19, Madrid-CARB-ES-19, and ICU-CARB-ES-19 groups, respectively, with *qnr*B1-like the most frequent in all groups (Table 4).

CP-Kpn COVID-19, ICU-CARB-ES-19, and Madrid-CARB-ES-19 isolates belonged to 20, 17, and 9 capsular polysaccharide *cps* loci, respectively (Appendix A). Regarding the COVID-19 group, 11 K-loci had more than one isolate and three K-loci, KL24 (27), KL102 (21), and KL122 (6), combined to include 64.3% of the isolates. High correlations between K-loci and STs were observed. KL24 isolates belonged to CC11 (13 isolates) and ST15 (14), all KL102 were ST307 (19) or ST3248 (2), and all KL122 were ST17. 

The yersiniabactin-encoding locus (*ybt*) alone was detected in 47.6% of COVID-19 isolates (40/84), mainly belonging to ST15 (14) and CC11 (14) with KL24 (all isolates of ST15 and 12 of CC11). Four different *ybt* lineages were detected with *ybt*10, associated with ICEKp4 and both to CC11 and ST15, the most frequent (31, 77.5%).

In the Madrid-CARB-ES-19 and ICU-CARB-ES-19 groups, the *ybt* locus alone was detected in 41.2% (14/34) and 37.5% (15/40) of the isolates, mainly belonging to ST11 and KL245 [6 (42.3%) and 5(33.3%), respectively]. The most frequent *ybt* lineage in both groups was *ybt*10, associated with ICEKp4 [10 (71.4%), and 6 (40%), respectively]. 

The other 44 CP-Kpn COVID-19, 20 Madrid-CARB-ES-19, and 25 ICU-CARB-ES-19 isolates were negative for all yersiniabactin (*ybt*), colibactin (*clb*), and aerobactin (*iuc*) loci (Appendix A).

Eight cases of bacteriemia due to CP-Kpn were detected in the 84 COVID-19 patients. Six isolates were ST307-KL102, one ST17-KL122 and one ST321-KL3, producing OXA-48 (*n* = 4), VIM-1 (*n* = 3) and KPC-3 (*n* = 1). ST307 isolates were associated to CTX-M-15 production. No major virulence markers were detected in the isolates causing bacteremia, and the *ybt* locus alone was detected in only one of these isolates.

### 2.5. Characterization of Plasmids Harboring the bla_VIM-1_ Gene

Due to the relatively high prevalence of VIM-1-producing isolates in the pandemic-period COVID-19 isolates, the plasmidID mapping tool was used to identify and reconstruct 14 plasmids harboring *bla*_VIM-1_ from COVID-19 isolates of CP-Kpn (4 ST307, 4 ST17, 3 ST485, and 3 ST321). In all cases, an IncL plasmid was identified, which was highly similar to NZ_CP023419.1 of 70,869 bp (average identity >95% with 98.2% average-coverage). A *bla*_VIM-1_-containing class 1 integron was detected in all plasmids, with the structure IS1-Int1-*bla*_VIM-1_-*aac*(6*′*)-*Ib*-*dfrB*1-*aadA*I-*catB*2-*qacEΔ*1*/sul*1*-*IS1326 (Figure 4). In addition, in all isolates belonging to ST485 and ST321, a c*atA*1 gene flanked by ISKpn14 and IS26 was detected upstream of the *bla*_VIM-1_ gene (Figure 5).

## 3. Discussion

During 2020, ICU admissions were significantly elevated because of the escalating global SARS-CoV-2 pandemic. According to several sources, up to 5% of patients infected with SARS-CoV-2 were admitted to the ICU [11,12]. Some ICUs operated at or above capacity for several months, which could have contributed to increased spread of opportunistic pathogens, such as *K. pneumoniae*, resulting from overcrowding and excessive demand on the healthcare system. In addition, the overuse of antimicrobials in ICUs is known to promote the development of MDR bacteria [13,14]. Retrospective observational research in Italy detected an increase from 6.7% to 50% in CPE in ICUs between 2019 and April 2020 [15], and other studies in medical centers in New York [16] and Italy [17] also documented increased detection of CPE in patients with COVID-19. Beyond the quantitative variations, it is also important to assess whether modifications to healthcare best practices, made out of necessity during the first waves of the COVID-19 pandemic, may have qualitatively influenced the prevalence or nature of CP-Kpn in COVID patients admitted to ICUs.

This study shows that the population of CP-Kpn affecting COVID-19 patients admitted to the ICUs of five hospitals in Madrid during the first year of the pandemic was mainly made up of OXA-48-producing isolates belonging to ST307. Comparison with a previous collection of CP-Kpn isolates in 2019 [5], just a year before the start of the pandemic, reveals a similar scenario. However, we note here certain evolutionary trends that deserve monitoring to determine whether they will consolidate or whether they reflect specific and/or local epidemiological situations. For example, it is worth noting that the number of VIM-1-producing *K. pneumoniae* isolates increased during the pandemic and that these isolates were associated with previously rare STs, including ST17, ST321, and ST485.

The present study is important because it reports WGS analysis of CP-Kpn isolates colonizing or infecting seriously ill COVID-19 patients at the height of the pandemic in a region and at a time when healthcare services were extremely stressed. Furthermore, WGS data on COVID-19 isolates of CP-Kpn were directly compared with a pre-pandemic Spanish national collection of representative CP-Kpn isolates collected a few months before the start of the pandemic. A limitation of the present study may be that different protocol designs were used to collect CP-Kpn strains during the pandemic and pre-pandemic periods; justified by the difficulty in collecting cases of infection/colonization by CP-Kpn during the first phase of the pandemic. 

Traditionally, and in general, VIM has been more frequent in Spain than in the rest of Europe. For example, in the European Survey on CPE (EuSCAPE) carried out in 2013–2014, 5.7% and 10.3% of all European CP-Kpn isolates and CP-Kpn from Spain, respectively, were VIM-producers [18]. However, the emergence and dissemination of OXA-48 as well as KPC and NDM has reduced the relative frequency of VIM-1 in Spain [5,19]. In the present study, a high frequency of VIM-1 linked to STs rarely associated with this type of carbapenemase (namely ST17, ST485, and ST307) has been detected in COVID-19 patients admitted to the ICU.

This work confirmed the predominance of the high-risk clone ST307 detected in previous studies [5,20]. However, in contrast to previous studies, four isolates in the present study carry the *bla*_VIM-1_ gene. The ST307 dispersion has been linked to the carbapenemases OXA-48 and KPC, with very infrequent previous reports of linkage between VIM-type carbapenemases and this ST [21,22].

ST17, ST485, and ST321 of *K. pneumoniae*, detected in this study carrying *bla*_VIM-1_ from COVID patients, are rare STs in CP-Kpn. Recently, the coexistence of *mcr*-1, *bla*_NDM-5_, and *bla*_CTX-M-55_ in a *K. pneumoniae* ST485 isolate was communicated [23], but there are no published reports of outbreaks due to carbapenemase-producing ST485. Regarding ST17 isolates, they were recently linked to the production of OXA-181 carbapenemase [22,24] and to sporadic cases of hypervirulent isolates [25,26], but they were not previously linked to production of VIM-1. In this study, no hypervirulent isolates were detected. Five VIM-1-producing ST321 isolates were recently detected in long-term care facilities in the Northern Italian region [27].

Relative to the pre-pandemic population diversity in CP-Kpn, the SDI was lower, indicating less diversity in the COVID-19 period than in the pre-COVID period, especially relative to ICU-CARB-ES-19 isolates of CP-Kpn. This fact could reflect the dissemination of intra-ICU-specific clones, facilitated by the large increase in ICU admissions during the pandemic [11,12]. 

Regarding the level of resistance to carbapenem antibiotics as the main target of carbapenemases, it should be noted that the profile of meropenem and/or imipenem susceptibility with ertapenem resistance was frequently detected in the three patient groups, with minor differences between them. This profile was conferred by and is mainly due to highly prevalent OXA-48 isolates of CP-Kpn [5,19], although it is also observed in patients with VIM-1-producing isolates [19,28]. It is worth noting the difference in susceptibility to meropenem (37.6%) and meropenem/vaborbactam (87.1%) in a collection in which the VIM-1- and OXA-48-producers predominated, carbapenemases that are not inhibited by vaborbactam. This fact is mainly because predominated isolates with meropenem MICs of 4-8 mg/L, and they were characterized as Susceptible, increased exposure (I); these MICs in the case of meropenem/vaborbactam were considered susceptible, according to EUCAST criteria. On the other hand, new antibiotics, such as cefiderocol, plazomicin, meropenem/vaborbactam, and imipenem/relebactam, have significantly improved the treatment options for CPE infections [13]. In our study, all CP-Kpn isolates showed > 73% susceptibility to these antibiotics with non-significant differences between the three groups. Minor differences among the three groups were mainly justified by the predominant type of carbapenemases in each group; and, specifically, different susceptibilities in aztreonam and ceftazidime/avibactam reflect different proportion of metallo-beta-lactamases by groups.

Modifications in the profile of antibiotic use in general, and in particular against infections produced by CP-Kpn, can contribute to the selection of different types of carbapenemases and clones that are prone to carry them. Both, the COVID-19 pandemic and the marketing of new antibiotics against this type of bacteria, could have been factors that may have contributed to change the use of antibiotics in ICUs [29,30].

Although chloramphenicol resistance genes were less frequent in pandemic-period isolates (40.5%) than in pre-pandemic isolates (70.5% and 70% in Madrid-CARB-ES-19 and ICU-CARB-ES-19 groups, respectively), no clear pattern was observed to justify this difference. In general, the absence of chloramphenicol resistance genes occurred mainly in OXA-48-producing strains of different STs in the three study groups. The *catB2* gene was mainly associated with VIM-1-producing isolates linked to a class 1 integron in COVID-19 patients (20.2%).

It is worth highlighting the presence of the same class 1 In624-like integron [31] and the IncL plasmid carrying VIM-1 in all emerging STs with this type of enzyme among CP-Kpn isolates in the COVID-19 group. Class 1 integron In624-like has been previously described in *Enterobacter cloacae* [31], *Citrobacter freundii* [32], and *Serratia marcescens* [33] in Spain, suggesting that it plays an important role in the interspecies transfer of *bla*_VIM-1_. 

IncL plasmids detected in this study are closely related to plasmids previously shown to be responsible for the worldwide spread of OXA-48 [34]. The main difference between the two is the presence of a class I integron with the *bla*_VIM-1_ carbapenemase in the former, instead of the *bla*_OXA-48_ transposon Tn1999 in the latter [33]. The great biological success of the IncL plasmid carrying OXA-48 [34] should alert us to the possible future dissemination of the IncL plasmid harboring VIM-1. This IncL-VIM-1 plasmid has been recently described in multidrug-resistant strains of *K. pneumoniae* [35] and *S. marcescens* [33] in Spain. The simultaneous presence of this plasmid in different emerging STs supports this hypothesis and requires surveillance.

High correlations between K-loci and STs have been described previously [5], including the predominant combinations KL24/ST11 and KL102/ST307.

Looking to the future, strategies to manage patients with COVID-19 should include approaches for mitigating the impact of MDR infections in this population. Updated knowledge about carbapenemase-producing Enterobacterales will allow for early detection of emerging mechanisms of resistance and the clones or mobile genetic elements that carry them and facilitate their spread. In this regard, the emergence of new VIM-1-producing *K. pneumoniae* clones linked to successful IncL plasmids and a class 1 integron, as reported here, is a matter of concern that requires continuous monitoring.

## 4. Materials and Methods

### 4.1. Study Design and CP-Kpn Isolates

This study was performed by the unrestricted national Spanish Antibiotic Resistance Surveillance Programme, operated by the official Spanish Public Health Institute (Instituto de Salud Carlos III). The study characterized 85 CP-Kpn isolates collected from individual COVID-19 patients in ICUs of five hospitals in Madrid. These isolates were collected between 1 April 2020 and 30 April 2021. For comparison, two groups of CP-Kpn isolates obtained before the COVID-19 pandemic began, during the CARB-ES-19 national multicenter study, were also analyzed. The first of these two pre-pandemic groups included 34 CP-Kpn isolates collected in Madrid, while the second pre-pandemic group included 40 CP-Kpn isolates from Spanish ICU patients obtained during CARB-ES-19. The overall results of the CARB-ES-19 study were recently published [5]. Briefly, 71 hospitals representing all 50 Spanish provinces participated in CARB-ES-19, whose goal was to collect the first ten non-duplicate consecutive isolates of carbapenem non-susceptible Kpn isolates from clinical samples from individual patients between February and May 2019. The CARB-ES-19 study included four hospitals in Madrid that collected 34 CP-Kpn isolates [5].

### 4.2. Drug Susceptibility Testing

Antibiotic susceptibility testing was performed using the broth microdilution susceptibility method (DKYMGN SensititreTM panels, Thermo Fisher Scientific, Waltham, MA, USA) [36]. Antibiotic gradient strips were used to study susceptibility to meropenem/vaborbactam and cefepime (bioMérieux, Marcy-l’Étoile, France), as well as imipenem/relebactam, plazomicin, and cefiderocol (Liofilchem, Roseto degli Abruzzi, Italy) in Mueller Hinton agar (bioMérieux, Marcy-l’Étoile, France). EUCAST v12.0 clinical breakpoints and guidelines for Enterobacterales were used to interpret the data. The Food and Drug Administration-approved susceptibility breakpoint of ≤2 mg/L was used for plazomicin.

### 4.3. Genomic Library Preparation and DNA Sequence Analysis

Genomic DNA paired-end libraries were generated using the Nextera XT DNA Sample Preparation Kit (Illumina, Inc., San Diego, CA, USA). These libraries were sequenced using the Illumina HiSeq 500 and NextSeq 500-500 high output v2.5 next-generation sequencers with 2 × 150 bp paired-end reads (Illumina, Inc.) Raw sequence data were submitted to the European Nucleotide Archive (PRJEB57245 and PRJEB50822 for isolates from COVID-19 patients and CARB-ES-19 isolates, respectively). Quality of short reads was assessed using FASTQC, and they were assembled into contigs with Unicycler 0.4.8 [37]. The quality of the assembly was assessed with QUAST (http://quast.bioinf.spbau.ru/, accessed on 3 December 2022). Prokka v1.14-beta [38] was used for automatic de novo assembly annotation.

### 4.4. Phylogenetic Analyses

STs were calculated according to multilocus sequence typing (MLST) schemes of the Institut Pasteur using Ariba v2.6.2 [39]. A simple diversity index (SDI) [40] was applied to analyze population diversity. A core genome multilocus sequence typing (cgMLST) that relies on species-specific schemes with a fixed number of chromosomal target genes was applied, consisting of 2538 *K. pneumoniae* targets provided by SeqSphere+ 3.5.0 (Ridom, Münster, Germany). A relatedness threshold of ≤10 alleles was applied for detecting related isolates, as recommended [10].

### 4.5. Analysis of Antimicrobial Resistance, Virulence Genes, and Capsular Locus

Antibiotic resistance genes were analyzed by Ariba v2-6.2 [39] using the CARD database (https://card.mcmaster.ca, accessed on 3 December 2022) and ResFinder (CGE server, https://cge.cbs.dtu.dk, accessed on 3 December 2022) with ID thresholds of 100% for β-lactamase variants and 98% for other genes. The K-locus and virulence genes were characterized using Kleborate [41]. The presence of *ybt*, *clb,* and *iuc* was used to assign a virulence score, as described previously [41].

### 4.6. Characterization of Plasmids Carrying Carbapenemase Genes

To reconstruct the plasmids carrying the *bla*_VIM_ genes, an in-house script (PlasmidID, https://github.com/BU-ISCIII/plasmidID, accessed on 3 December 2022) was used. The aims were to (i) map reads over a curated plasmid database, to find those with higher coverage and to assemble these reads de novo; (ii) make local alignments to localize resistance and replicative genes; and (iii) generate a graphic representation of the plasmids identified.

## Figures and Tables

**Figure 1 antibiotics-12-00107-f001:**
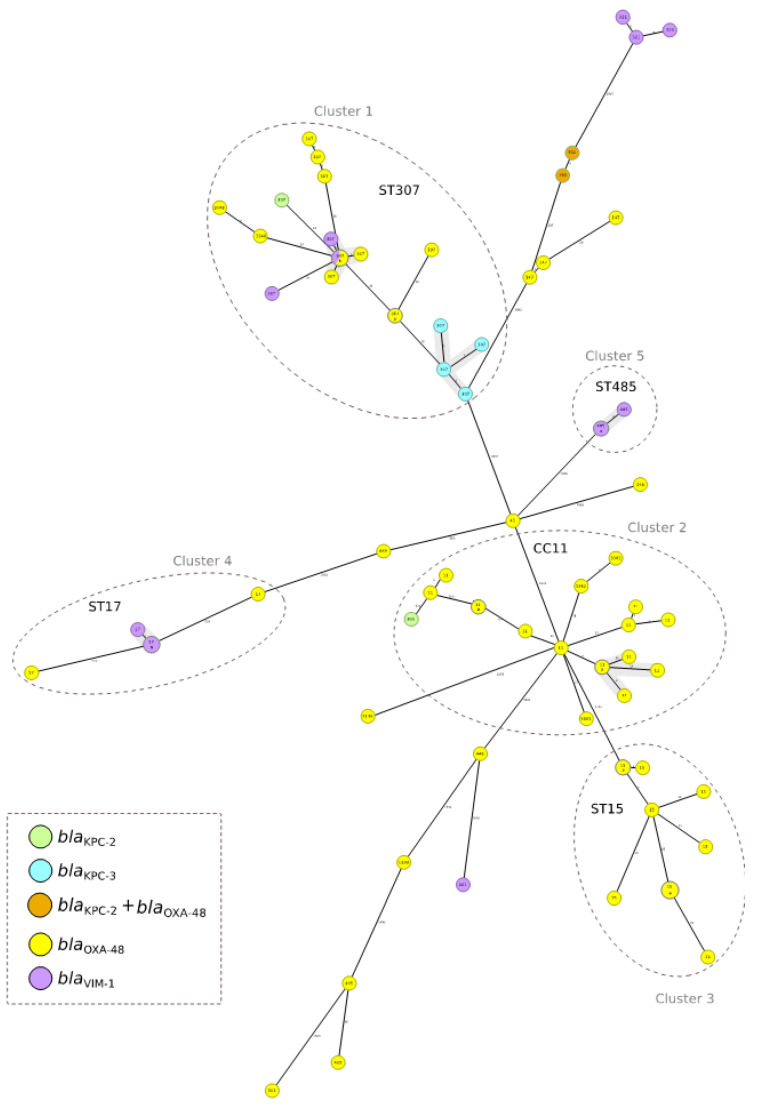
Population structure of carbapenemase-producing *Klebsiella pneumoniae* isolated from COVID patients: Distances shown in minimum-spanning tree are based on cgMLST of 2358 genes using the parameter ‘pairwise ignoring missing values’. Colors in each circle indicate carbapenemase type and number indicate the sequence type (ST). Grey ovals represent clusters. Where a circle corresponds to more than one isolate, the number of isolates is indicated in bold font. Gray shadows represent groups of strains; a threshold of 10 alleles was applied.

**Figure 2 antibiotics-12-00107-f002:**
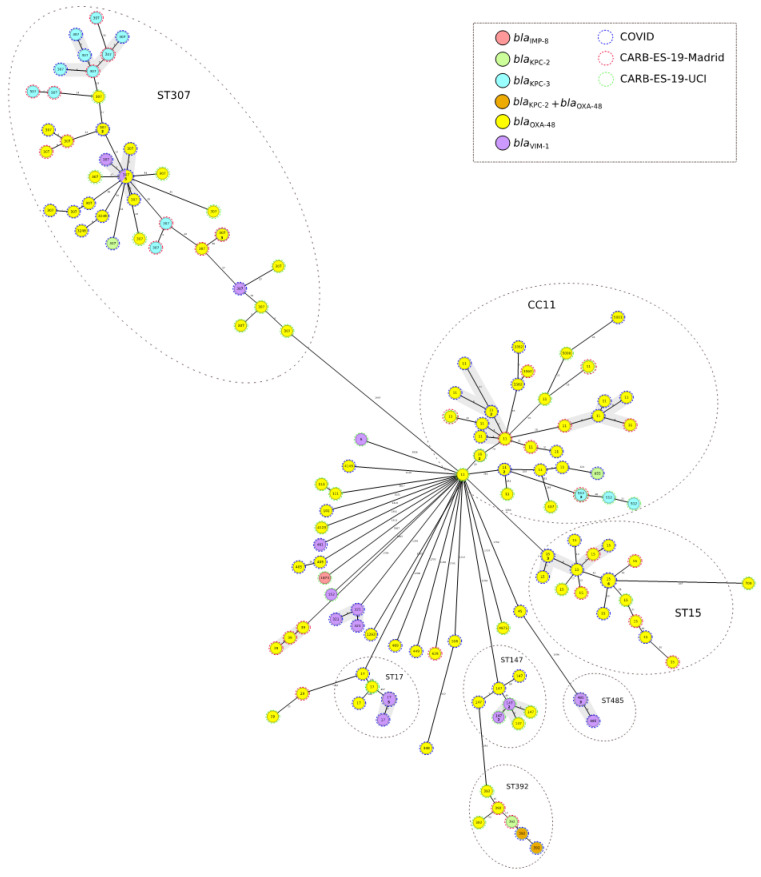
Population structure of carbapenemase-producing *Klebsiella pneumoniae* isolated from COVID patients compared to pre-pandemic isolates: Distances shown in minimum-spanning tree are based on cgMLST of 2358 genes using the parameter ‘pairwise ignoring missing values’. Colors in each circle indicate carbapenemase type, and numbers indicate the sequence type (ST). Dotted circles in blue, red, and green represent the three study groups (COVID-19, Madrid-CARB-ES-19, ICU-CARB-ES-19). Grey ovals represent clusters. For circles that correspond to more than one isolate, the number of isolates is indicated in bold font. Gray shadows represent groups of strains; a threshold of 10 alleles was applied.

**Figure 3 antibiotics-12-00107-f003:**
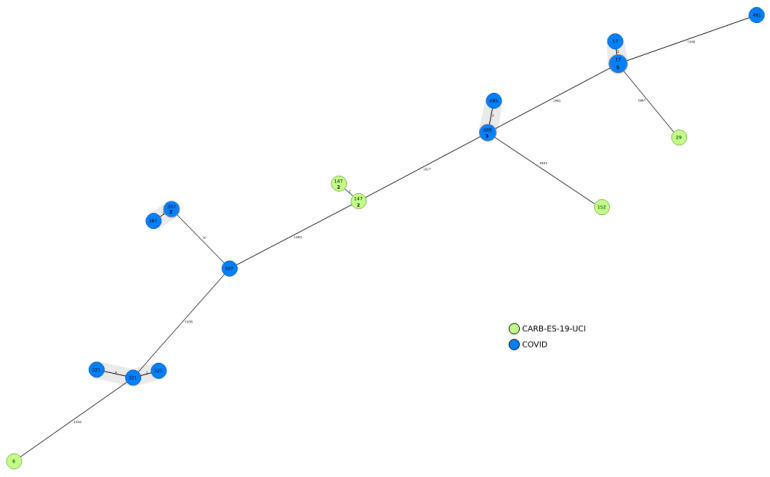
Population structure of VIM-1-producing *Klebsiella pneumoniae* in this study: Distances shown in minimum-spanning tree are based on cgMLST of 2358 genes using the parameter ‘pairwise ignoring missing values’. Colors in each circle indicate the three study groups (COVID-19, Madrid-CARB-ES-19, ICU-CARB-ES-19) and numbers indicate the sequence type (ST). For circles that correspond to more than one isolate, the number of isolates is indicated in bold font.

**Figure 4 antibiotics-12-00107-f004:**
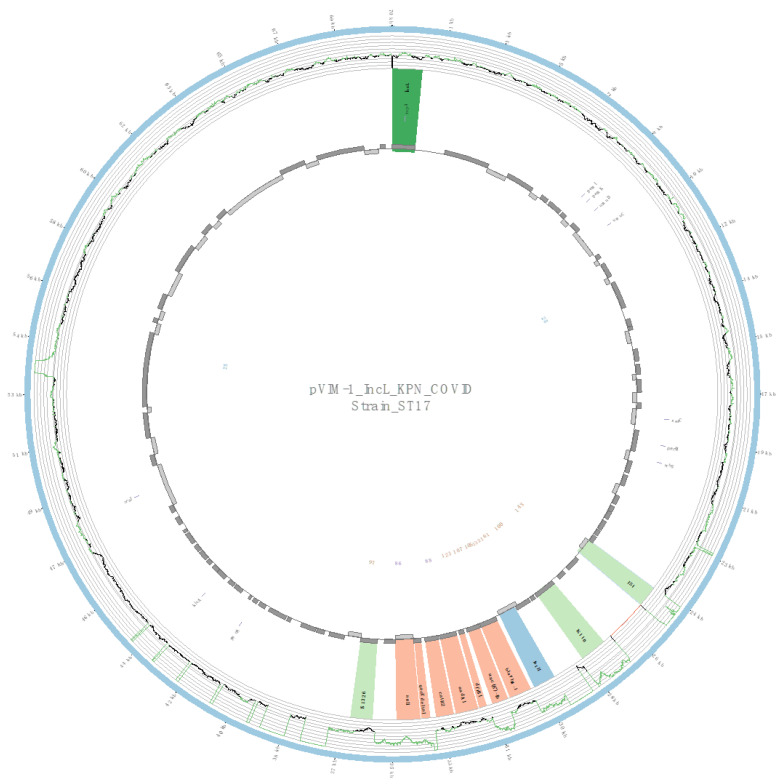
Overview of the IncL plasmid harboring *bla*_VIM-1_ detected in a strain of ST17: The figure represents the homology between the IncL plasmid and a highly similar plasmid identified in the GenBank database (blue outer ring). The graph represents the reads mapped against this reference sequence with a depth of coverage ranging from 0 (red) to 500, with orange indicating values of 1 to 20 reads and green indicating values higher than 200 reads. Gray boxes represent the coding sequence from automatic annotation, with dark and light colors being used when they were found on the forward or the reverse strand, respectively. Colored stripes represent a more detailed annotation that includes antibiotic resistance genes in red, insertion sequences (IS) in light green, integrases in blue, and Rep genes in dark green. The homology between the reference plasmid and the assembled contigs is represented in the inner ring, with each contig colored according to its number.

**Figure 5 antibiotics-12-00107-f005:**
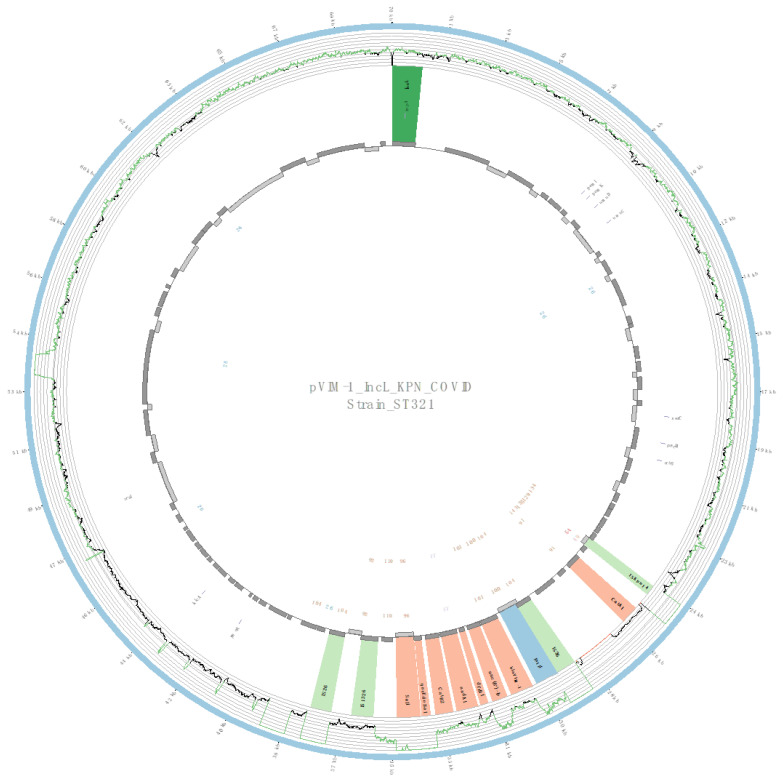
Overview of the IncL plasmid harboring *bla*_VIM-1_ detected in a strain of ST321: The figure represents the homology between the IncL plasmid and a highly similar plasmid identified in the GenBank database (blue outer ring). The graph represents the reads mapped against this reference sequence with a depth of coverage ranging from 0 (red) to 500, with orange indicating values of 1 to 20 reads and green indicating values higher than 200 reads. Gray boxes represent the coding sequence from automatic annotation, with dark and light colors being used when they were found on the forward or the reverse strand, respectively. Colored stripes represent a more detailed annotation that includes antibiotic resistance genes in red, insertion sequences (IS) in light green, integrases in blue, and Rep genes in dark green. The homology between the reference plasmid and the assembled contigs is represented in the inner ring, with each contig colored according to its number.

**Table 1 antibiotics-12-00107-t001:** Antibiotic susceptibility of different groups of carbapenemase-producing *Klebsiella pneumoniae* isolates as determined by the microdilution method and antibiotic gradient strips according to EUCAST (*) and FDA (**) clinical breakpoints, respectively.

	CP-kpn COVID-19 (*n* = 85)					CP-kpn CARB-ES-19 Madrid (*n* = 34)					CP-kpn CARB-ES-19 UCIs (*n* = 40)				
Antibiotics	S (%)	R (%)	MIC50 ^a^	MIC90 ^a^	Range ^a^	S (%)	R (%)	MIC50 ^a^	MIC90 ^a^	Range ^a^	S (%)	R (%)	MIC50^a^	MIC90 ^a^	Range ^a^
Cefiderocol *	98.8	1.2	0.064	0.64	≤0.015 to 4	97.3	2.9	0.06	0.25	≤0.015 to 4	100	0	0.12	0.5	≤0.015 to 1
Plazomicin **	97.6	1.2	0.5	1	0.125 to 8	100	0	1	2	0.25 to 2	97.5	2.5	1	1	0.5 to 8
Colistin	85.9	14.1	1	>8	0.5 to 8	91.2	8.8	1	2	0.5 to >8	85	15	1	>8	0.5 to >8
Meropenem/vaborbactam *	87.1	12.9	1	64	0.032 to >64	82.4	17.6	1	16	0.015 to 128	92.5	7.5	0.5	8	0.03 to >64
Ceftazidime/avibactam	78.8	21.2	2	>16	≤0.5 to >16	100	0	2	8	≤0.5 to 8	72.5	27.5	2	>16	≤0.5 to >16
Imipenem/relebactam *	82.4	17.6	1	>32	0.125 to >32	73.5	26.5	1	>32	0.5 to >32	77.5	22.5	1	32	0.25 to >32
Amikacin	80	11.8	≤4	32	4 to >32	76.5	23.5	8	16	≤4 to >32	72.5	27.5	8	32	≤4 to >32
Imipenem	41.2	28.2	4	16	≤0.5 to >16	52.9	35.3	2	>16	≤0.5 to >16	57.5	35	2	>16	1 to >16
Meropenem	37.6	22.4	2	>16	0.5 to >16	52.9	20.6	2	>16	0.5 to >16	50	32.5	4	>16	0.5 to >16
Gentamicin	43.5	49.4	4	>8	≤0.5 to >8	32.3	67.6	>8	>8	≤0.5 to >8	40	60	>8	>8	≤0.5 to >8
Cotrimoxazole	20	76.5	>8	>8	≤1 to >8	17.6	82.4	>8	>8	≤1 to >8	32.5	67.5	>8	>8	≤1 to >8
Tobramycin	24.7	75.3	>8	>8	≤1 to >8	17.6	82.4	>8	>8	≤1 to >8	27.5	72.5	>8	>8	≤1 to >8
Aztreonam	14.1	84.7	>32	>32	0.5 to >32	8.8	91.2	>32	>32	≤0.5 to >32	17.5	75	>32	>32	≤0.5 to >32
Cefepime *	4.7	82.4	16	>256	0.025 to >256	8.8	85.3	64	>256	0.25 to >256	15	85	32	>256	0.25 to >256
Ceftazidime	4.7	92.9	>16	>16	≤0.5 to >16	2.9	91.2	>16	>16	1 to >16	17.5	82.5	>16	>16	≤0.5 to >16
Ceftolozane/tazobactam	4.7	95.3	>32	>32	1 to >32	2.9	97.1	>32	>32	2 to >32	12.5	87.5	>32	>32	≤0.5 to >32
Cefotaxime	1.2	95.3	>8	>8	1 to >8	0	97.1	>8	>8	2 to >8	5	85	32	>256	0.25 to >256
Ciprofloxacin	7.1	92.9	>2	>2	≤0.06 to >2	0	100	>2	>2	>2	10	87.5	>2	>2	≤0.06 to >2
Ertapenem	2.4	96.5	>2	>2	0.5 to >2	0	100	>2	>2	1 to >2	2.5	97.5	>2	>2	0.5 to >2

S: susceptible, standard dosing regimen. R: resistant. MIC: minimum inhibitory concentrations. ^a^ Expressed in mg/L. * Antimicrobial susceptibility determined by antibiotics gradients strips according to EUCAST clinical breakpoints. ** Antimicrobial susceptibility determined by antibiotics gradients strips according to FDA breakpoints.

**Table 2 antibiotics-12-00107-t002:** Antibiotic susceptibility according to carbapenemase groups in CP-kpn COVID-19 isolates.

	Susceptibility (%) in CP-kpn COVID-19		
Antibiotics	OXA-48-Group Producing Isolates (*n* = 59)	KPC-Group Producing Isolates (*n* = 6)	VIM-Group Producing Isolates (*n* = 18)
Cefiderocol *	98.3	100	100
Plazomicin **	96.6	100	100
Colistin	86.4	66.7	100
Meropenem/vaborbactam *	88.1	100	77.8
Ceftazidime/avibactam	100	100	0
Imipenem/relebactam *	88.1	100	55.6
Amikacin	84.7	66.7	66.7
Imipenem	57.6	0	5.6
Meropenem	49.2	0	16.7
Gentamicin	50.8	50	22.2
Cotrimoxazole	25.4	33.3	0
Tobramycin	32.2	33.3	0
Aztreonam	6.8	0	44.8
Cefepime *	6.8	0	0
Ceftazidime	6.8	0	0
Ceftolozane/tazobactam	6.8	0	0
Cefotaxime	1.7	0	0
Ciprofloxacin	8.5	0	5.6
Ertapenem	0	0	11.1

* Antimicrobial susceptibility determined by antibiotics gradients strips according to EUCAST clinical breakpoints. ** Antimicrobial susceptibility determined by antibiotics gradients strips according to FDA breakpoints. Isolates with two carbapenemases of different groups are excluded (*n* = 2).

**Table 3 antibiotics-12-00107-t003:** Comparative data of the population structure of carbapenemase-producing *K. pneumoniae* from COVID-19 patients in Madrid ICUs (CP-Kpn COVID-19) versus two groups of prepandemic isolates (CARB-ES-19 project) from patients in Madrid (CP-Kpn Madrid-CARB-ES-19) and from Spanish patients admitted in ICUs (CP-Kpn ICU-CARB-ES-19).

	CP-Kpn COVID-19	CP-Kpn Madrid- CARB-ES-19	CP-Kpn ICU- CARB-ES-19
Number of isolates	84	34	40
Number of STs	20	9	18
Average of isolates per ST (range)	4.2 (1–20)	3.8 (1–13)	2.2 (1–9)
Number of single isolates by ST (%)	9 (10.7)	3 (8.8)	11 (27.5)
SDI	23.8	26.5	45
OXA-48 (%)	60 (71.4)	25 (73.5)	27 (67.5)
KPC (%)	8 (9.5)	10 (29.4)	5 (12.5)
NDM (%)	0	0	0
VIM (%)	18 (21.4)	0	7 (17.5)
IMP (%)	0	0	1 (2.5)
ST307; *n* (%); carb. types	20 (23.8); OXA-48 (11), VIM-1 (4), KPC-3 (4), KPC-2 (1)	13 (38.2); KPC-3 (7), OXA-48 (6)	9 (22.5); OXA-48 (8), KPC-3 (1)
ST15; *n* (%); carb. types	15 (17.8); OXA-48	5 (14.7); OXA-48	2 (5); OXA-48 (2)
ST11; *n* (%); carb. types	14 (16.6); OXA-48	6 (17.6); OXA-48	7 (17.5); OXA-48 (6), KPC-2 (1)
ST17; *n* (%); carb. types	8 (9.5); VIM-1 (6), OXA-48 (2)	0	1 (2.5); OXA-48
ST147; *n* (%); carb. types	3 (3.6); OXA-48	0	4 (10); VIM-1
ST512; *n* (%); carb. types	0	2 (5.9); KPC-3	3 (7.5); KPC-3
ST485; *n* (%); carb. types	4 (4.8); VIM-1	0	0
ST39; *n* (%); carb. types	0	3 (8.8); (OXA-48)	0

Carb. Types: Carbapenemase types. SDI: Simple Diversity Index. ST: Sequence Type.

**Table 4 antibiotics-12-00107-t004:** Identification of the most frequent antibiotic resistance genes found in carbapenemase-producing *K. pneumoniae* from COVID-19 patients in Madrid ICUs (CP-Kpn COVID-19) versus two groups of prepandemic isolates (CARB-ES-19 project) from patients in Madrid (CP-Kpn Madrid-CARB-ES-19) and from Spanish patients admitted in ICUs (CP-Kpn ICU-CARB-ES-19).

Type of Resistance	Resistance Genes	% CP-kpn COVID-19 (*n* = 85)	% CP-kpn CARB-ES-19 Madrid (*n* = 34)	% CP-kpn CARB-ES-19 UCIs (*n* = 40)
ESBL	*bla* _CTX-M-15_	76.1	76.5	52.5
	*bla* _CTX-M-9_	2.4	0	5
	*bla* _SHV-12_	2.4	0	5
	*bla* _SHV-2_	1.2	0	0
	*bla* _CTX-M-65_	0	0	2.5
Aminoglycoside	*aac*(6*′*)-*Ib-cr*	50	64.7	45
	*aac(*3*)-IIa*	33.3	58.8	42.5
	*aac(*6*′)-Ib*	19	2.9	22.5
	*aph*	70.2	82.3	55
	*aph* (3``)-*Ib/aph* (6)-*Id* *	53.6	67.6	42.5
Chloramphenicol	*cat*	40.5	70.5	70
	*catA*1 *	22.6	0	0
	*catB*3 *	0	67.7	42.5
Sulfonamides	*sul*	75	76.4	72.5
	*sul*1	42.8	14.7	42.5
	*sul*2	55.9	67.6	40
Trimethoprim	*dfr*	86.9	82.3	65
	*dfrA*14 *	44	67.5	42.5
Tetracyclines	*tet*	42.8	26.5	32.5
	*tetA* *	36.9	26.5	30
Colistin	*mcr*	2.4	0	5
	*mcr-*9 *	2.4	0	5
Quinolone	*qnr-like*	45.2	61.8	50
	*qnrB1-like* *	22.6	61.8	42.5

ESBL: extended-spectrum β-lactamase. * Only the most frequent variants found in our study population are detailed.

## Data Availability

The datasets presented in this study can be found in the European Nucleotide Archive (PRJEB57245 and PRJEB50822 for isolates from COVID-19 patients and CARB-ES-19 isolates, respectively).

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
