# Peer review of "Carbapenemase-Producing Klebsiella pneumoniae in COVID-19 Intensive Care Patients: Identification of IncL-VIM-1 Plasmid in Previously Non-Predominant Sequence Types"

_antibiotics, 2023, doi:10.3390/antibiotics12010107_

Round 1

Reviewer 1 Report

Cañada-García et al. analyzed the molecular characteristics of 85 strains of carbapenemase-producing K. pneumoniae isolated from clinical samples or surveillance swabs among COVID-19 patients admitted to ICUs of five hospitals in Madrid. The virulome and resistome analysis, conducted through WGS, and the sequence typing were than compared to those of 74 pre-pandemic isolates to describe potential changes in molecular epidemiology of carbapemem-resistant Enterobacterales occurred during the pandemic.

The manuscript is well written, and the topic is interesting. However, there are some points that the authors should address:

1.      In my opinion it would be useful to report the antimicrobial susceptibility rates stratified according to the type of carbapenemase produced to correlate the genotypic and phenotypic characteristics of isolates.

2.      It is somewhat surprising that group of strains producing mostly OXA-48 and metallo-beta-lactamases (in about 70% and 20% of cases, respectively), shows a susceptibility rate of almost 90% to meropenem-vaborbactam, considering that vaborbactam has demonstrated no significant in vitro activity against class B and class D carbapenemases. The authors should address this point in the discussion.

3.      If possible, the authors should report the proportion of infected patients that were previously colonized by the same strain.

4.      Finally, it may be interesting to have data on the clinical severity of infections, in order to correlate the virulome analysis and sequence typing with the clinical characteristics of patients.  

Reviewer 2 Report

This study doesn't address the use of antibiotics in the patients from whom the samples were collected and how it could lead to the enrichment of certain specific genotypes which may or may not complicate the results. I want the authors to acknowledge or address this aspect in the study as a limitation or argue against it. 

Round 2

Reviewer 2 Report

None